# An Optimized Hybrid Approach for Feature Selection Based on Chi-Square and Particle Swarm Optimization Algorithms

**Amani Abdo** [1,2,*] , **Rasha Mostafa** [2,*] and **Laila Abdel-Hamid** [2]

1   Faculty of Computing, Arab Open University, El-Shorouk 51, Cairo 11211, Egypt
2   Faculty of Computers & Artificial Intelligence, Helwan University (HU), Ain Helwan, Cairo 11795, Egypt; laila.abdelhamid@fci.helwan.edu.eg
*   Correspondence: amani.abdo@aou.edu.eg (A.A.); rashamohamed_psw@fci.helwan.edu.eg (R.M.)

**Abstract:** Feature selection is a significant issue in the machine learning process. Most datasets include features that are not needed for the problem being studied. These irrelevant features reduce both the efficiency and accuracy of the algorithm. It is possible to think about feature selection as an optimization problem. Swarm intelligence algorithms are promising techniques for solving this problem. This research paper presents a hybrid approach for tackling the problem of feature selection. A filter method (chi-square) and two wrapper swarm intelligence algorithms (grey wolf optimization (GWO) and particle swarm optimization (PSO)) are used in two different techniques to improve feature selection accuracy and system execution time. The performance of the two phases of the proposed approach is assessed using two distinct datasets. The results show that PSOGWO yields a maximum accuracy boost of 95.3%, while chi2-PSOGWO yields a maximum accuracy improvement of 95.961% for feature selection. The experimental results show that the proposed approach performs better than the compared approaches.

**Keywords:** feature selection; artificial intelligence; chi-square; particle swarm optimization; grey wolf optimization; swarm intelligence

## 1. Introduction

Feature selection is a process that seeks to discover and remove features from a dataset that are not relevant or useful. These features are often perceived as unnecessary or extraneous to the problem being analyzed. Feature selection is used to generate a subset of attributes to use in constructing models for classification purposes [1]. Feature selection has been applied in a range of intelligent and expert systems such as intrusion detection [2], cancer detection [3], sentiment analysis [4], and disease detection and classification [5].

Feature selection methods can be categorized into wrapper-based methods, filter-based methods, and hybrid-based methods that combine elements from both approaches [6,7]. Filter-based methods (such as information gain [8], chi-square [9], minimum redundancy maximum relevance (MRMR) [1]) use statistical methods to rank and select the most pertinent features. This technique is applied prior to running the machine learning classifier and does not interact directly with it [10]. Wrapper-based methods utilize an optimization algorithm in conjunction with the classifier to identify the most suitable features. Wrapper-based methods are typically employed for feature selection because of their efficiency in decreasing the amount of features and increasing the classifier's accuracy, as they have a direct connection to the classifier being used [10]. Wrapper-based methods are slower and more computationally expensive than filter-based feature selection techniques. Hybrid methods combine two distinct methods in order to reap the benefits of both (e.g., ECCSP-SOA [11], CS-BPSO [12], ISSA [10]).

Metaheuristic algorithms are employed in feature selection techniques to reduce computational complexity. These algorithms efficiently and accurately optimize feature

selection problems. Swarm intelligence (SI) and evolutionary algorithms (EA) are the two primary categories into which these algorithms can be classified.

I.   An evolutionary algorithm takes advantage of processes such as reproduction, mutation, recombination, and selection, which are modelled after biological evolution. The fitness function identifies the quality of candidate solutions to the optimization problem, which act as members of a population. The original population changes after several iterations of the evolutionary algorithm, moving towards global optimization [13].

II.  Swarm intelligence: The foundation of swarm intelligence is self-organizing group behavior, which involves the intelligence that is generated by the collective contributions of numerous individuals. Since these connections, as seen in nature like in bees or ants, do not exist naturally in humans, technology uses swarm artificial intelligence (AI) to provide feedback to human members as demonstrated in [14,15]. Collective behavior shows that united systems do better than the majority of single individuals. Ad hoc data and sharing are dynamically generated by the group, and the basis of agreement is the dissemination of collective wisdom. Briefly stated, swarm intelligence relies on the "knowledge of the public", and is desperately needed to address a myriad of questions [16].

One of the most popular swarm intelligence techniques is the particle swarm intelligence algorithm (PSO). Comparing PSO to other metaheuristic algorithms like GA and genetic programming (GP), it has been demonstrated that PSO is computationally less expensive and can converge more quickly. Additionally, PSO tends to be easy to implement. In terms of speed and memory requirements, it is computationally cheap and has fewer adjustable parameters [17]. PSO can be customized to perform feature subset selection, aiming to find the optimal combination of features that should be included in the model. This process is essential for enhancing model performance by reducing the dimensionality of the data and eliminating irrelevant or redundant features. Therefore, PSO has been used as an effective technique in many fields, including feature selection [18]. The grey wolf optimizer (GWO) is one of the most recent and popular swarm intelligence algorithms. Compared to other swarm intelligent optimization methods, GWO offers the following benefits: no parameters to change, easy to implement and adapt for optimization challenges, adaptability, and scalability. GWO has been widely used as a feature selection approach in several fields during the past few years, including intrusion detection [19], big data analytics [20], and image classification [21].

This paper is organized as follows: recent research on feature selection is included in Section 2. Section 3 presents the complete chi2-PSOGWO feature selection technique, encompassing all its phases. A background study of the algorithms used is provided in Section 4. Section 5 presents the experimental results obtained, while a summary of and observations derived from the experiments are presented in Section 6. Section 7 comprises the conclusion and outlines future work.

## 2. Related Works

There are several attempts in the literature to develop swarm intelligence algorithms to improve the process of feature selection and achieve best practice in machine learning.

Seyyedabbasi, Amir [22] applied a wrapper feature selection algorithm to biological data. A binary sand cat swarm optimization algorithm was proposed to solve the local optima problem, which caused more complexity, execution time, and cost. They applied their experiment on 10 datasets. They achieved an average accuracy of 91.18% over all datasets.

Zivkovic, Miodrag et al. [23] suggested an improvement to the original SSA algorithm. They observed that the drawbacks of the original SSA are insufficient exploration, average exploitation power (conditional drawback), and the intensification–diversification trade-off. The modification was on the current best solution (F). Two control parameters were introduced instead of random ones. They combined using the original follower equation of

SSA with the nominal equation of SCA for followers. Their proposed algorithm inherited all of the original SSA's complexity but outperformed it in terms of accuracy for most datasets.

Adamu, Abdullahi [11] proposed an enhanced chaotic crow search and particle swarm optimization technique (ECCSPSOA) and offered a hybrid binary version to address feature selection issues. The ECCSPSOA approach merely targeted only a few selected crows with the best food to improve the performance of the original CSA's random following of every crow. Another enhancement was the use of chaotic sequences with a starting value of 0.7 to replace the random variables in the PSO and CSA. The computational results showed that their method had an average accuracy of 89.67%.

BinSaeedan and Alramlawi [12] suggested a hybrid feature selection algorithm that combined binary particle swarm optimization (BPSO) and chi-square BPSO (CS-BPSO) to increase the efficiency of Arabic email authorship analysis. They considered both dynamic and static features. The outcomes demonstrated that the CS-BPSO approach used dynamic features to attain remarkable results. They achieved better results with small datasets, short texts, and imbalanced datasets. In terms of accuracy, the SVM model performed up to 94.32% for static features and 98.84% for dynamic features, but the KNN model had a better f1-score in AA with dynamic features, coming in at 0.97.

Zouache and Abdelaziz [24] presented a cooperative swarm intelligence algorithm based on quantum computation concepts and rough set theory (QCSIA-FS) for feature selection. Particle swarm optimization (PSO) and a firefly system (FA) were combined in the algorithm to reduce its complexity and improve classification performance. An average accuracy of 86.79% for the CART classifier and 86.94 for the KNN classifier were obtained.

Wang, Wu [14] introduced a feature selection strategy based on a modified ant lion optimizer (MALO) and WSVM to learn the dimensionality of hyperspectral image HSIs. The classification accuracy for the Botswana and KSC datasets was 93.98% and 93.45%, respectively, when using the MALO algorithm. The experimental results revealed that their method achieved a satisfactory classification accuracy by utilizing fewer bands and exhibiting a reasonable convergence orientation.

Sheykhizadeh and Naseri [25] presented a swarm intelligence metaheuristic invasive weed optimization. Four separate experimental datasets with NIR and FTIR spectral information were investigated for this purpose. The outcomes demonstrated that IWO's performance was on par with that of PSO, GA, and ACO techniques.

Chen, Zhou [26] developed a spiral-shaped mechanism (HPSO-SSM) to choose the best features for classification using a wrapper-based technique and a hybrid particle swarm optimization. They added three improvements to HPSO-SSM. According to the applicable datasets, the experimental results demonstrated that their algorithm's average accuracy was 95.07.

Mahapatra, Majhi [1] used a two-stage hybrid model, MRMR-SSA. The first stage used a filter-based approach to remove unnecessary and unrelated features. The extracted features were subsequently fed into the wrapper method during the second stage, where the salp swarm algorithm (SSA) was employed. Their analysis demonstrated that their method produced improved outcomes, with an accuracy rate ratio of 96.60%.

Tubishat, Idris [10] suggested an enhanced version of the SSA algorithm that addressed feature selection issues utilizing the OBL technique and the LSA algorithm. The enhancements, which were integrated into the standard SSA, were utilized to prevent the SSA from being trapped in local optimum conditions and increase population diversity. To increase the diversity of the SSA population, the ISSA received the advantage of the OBL method. Additionally, a new LSA algorithm was incorporated into the ISSA to prevent it from getting stuck in local optimums. They achieved a maximum accuracy of 99.4%. A drawback of the ISSA is that it chooses more features than other optimization techniques across four of the eighteen employed datasets.

Mostafa, R.R., et al. [27] proposed a chameleon swarm algorithm with a consumption AEO operator as a new iteration of the chameleon swarm algorithm (CSA) for feature selection. They proposed three modifications to enhance the performance of the original

CSA, aiming to improve the harmony between exploitation and exploration. First, a nonlinear transfer operator was suggested. Then, to prevent stagnation and early convergence, they included a randomization Lévy flight control parameter. Thirdly, they improved the artificial ecosystem-based optimization (AEO) algorithm's consumption operator, which strengthens the original CSA's global search approach. Compared to their rivals, they increased the speed of convergence towards the best solution for breast cancer detection. The drawbacks of that algorithm were the large computational time and the large number of selected features compared with their competitors.

El-Kenawy and Eid [28] introduced a hybrid optimizer optimization procedure that started with a random sample of individuals. Such individuals had suppressed potential solutions to the issue at hand. Alpha, beta, and delta represented the first three leaders, identified after evaluating the fitness function for each individual during each iteration. After that, the population was evenly divided into two classes, the first of which adhered to GWO processes and the other to PSO procedures. The PSO and GWO algorithms were utilized to identify and target potentially significant areas in the search space, resulting in a comprehensive exploration of these regions.

Alrefai and Ibrahim [29] used microarray datasets as the basis for combining an ensemble learning method with particle swarm optimization for feature selection and cancer classification. The preliminary result indicated that the performance results for colon cancer, breast cancer, leukemia, ovarian cancer, and central nervous system cancer, respectively, were 92.86%, 86.36%, 100%, 100%, and 85.71% in terms of accuracy.

## 3. Background

### 3.1. Overview of PSO

As shown in Algorithm 1, PSO models how knowledge of social behavior grows over time and how groups communicate when exchanging private knowledge about migratory patterns, flocking, or hunting. They are known as a swarm and particles, respectively, and together, they make up a solution. Using its own and its neighbors' information, a particle changes its position.

---

**Algorithm 1.** PSO pseudo code:

1: **initialize** population of particles and velocities
2: **while** t < maximum number of iterations
3:       **calculate** the fitness of all particles
4       **updating** position and fitness of particles
5:    **choose** the particle of best fitness value and the Gbest of all particles
6:    **for** each particle
7:          **calculate** the velocity of particle by Equation (2)
8:       **update** particle position by Equation (1)
9:    **end for**
10: **End while**

---

The swarm starts by producing a collection of random particles, along with their positions and velocities. Equations (1) and (2) represent the method that is used to update the particles' positions:

$$x_{ij}^{(t+1)} = x_{ij}^{(t)} + v_{ij}^{(t+1)} \tag{1}$$

$$v_{ij}^{(t+1)} = wv_{ij}^{(t)} + c1r1(x_{ij}^{p(t)} - x_{ij}^{(t)}) + c2r2(x_{ij}^{g(t)} - x_{ij}^{t}) \tag{2}$$

where $t$ is the current iteration and $w$ denotes an inertia weight and is used to speed up population convergence. When $x_{ij}$ is the i-th particle location in the $j$-th dimension, and $v_{ij}$ is the $i$-th velocity in the $j$-th dimension. Acceleration coefficients are expressed by the constants $c1$ and $c2$. The terms $x_{ij}^{p(t)}$ and $x_{ij}^{g(t)}$ denote particle i's best prior position in the $j$-th dimension, respectively. $r1$ and $r2$ are random parameters between 0 and 1.

Afterwards, each particle is assessed by PSO's main loop using a fitness function, and the results are checked against the local best and global best values [30].

### 3.2. Overview of GWO

The leadership organization and hunting tactics of grey wolves are modelled by the GWO algorithm. Grey wolf packs named alpha, beta, delta, and omega are used to imitate the leadership structure. The three essential elements of hunting, looking for prey, surrounding prey, and attacking prey, are also utilized. The pseudocode of GWO algorithm is presented in Algorithm 2.

---

**Algorithm 2.** GWO pseudo code:

---

1: **initialize** grey wolf populations
2: **initialize** a, A and c values
3: **calculate** the fitness of each search agent
4: $x_\alpha$ = The best search agent
   $x_\beta$ = The second best search agent
   $x_\delta$ = The Third best search agent
5: **while** t < maximum number of iterations
6:     **for** each GWO search agent
7:         update the position of current search agent by Equation (5)
8: **end for**
9: **update** A, c, w
10:            calculate the fitness of all search agents
11:     **update** $x_\alpha$, $x_\beta$, $x_\delta$
12: **End while**

---

The grey wolf hunting technique can be summarized as follows: It is reasonable to assume that the alpha (the best candidate solution), beta, and delta have the greatest understanding about prospective prey locations. In order to force the other search agents, including the omegas, to update their locations in accordance with the positions of the best search agents, the first three best answers are kept. Grey wolves update their locations using the following equations [31].

$$D_\alpha = |\vec{c_1} \cdot \vec{x_\alpha} - \vec{x}|, \; D_\beta = |\vec{c_2} \cdot \vec{x_\beta} - \vec{x}|, \; D_\delta = |\vec{c_3} \cdot \vec{x_\delta} - \vec{x}| \tag{3}$$

$$\vec{x_1} = \vec{x_\alpha} - \vec{A_1} \cdot \vec{D_\alpha} \, , \; \vec{x_2} = \vec{x_\beta} - \vec{A_2} \cdot \vec{D_\beta} \, , \; \vec{x_3} = \vec{x_\delta} - \vec{A_3} \cdot \vec{D_3} \tag{4}$$

$$\overrightarrow{x(t+1)} = \frac{\vec{x_1} + \vec{x_2} + \vec{x_3}}{3} \tag{5}$$

where $\vec{x_1}$, $\vec{x_2}$, $\vec{x_3}$ are the distances between each $\delta$, $\beta$, and $\alpha$ and the prey. t indicates the current iteration and $A$, $C$ are coefficient vectors given by $\vec{A} = 2\vec{a} \cdot \vec{r_1} - \vec{a}$, $\vec{c} = 2 \, \vec{r_2}$.

### 3.3. Overview of Chi-Square

The chi-square score of each feature and target is computed for feature selection $\chi^2$, and the top two features are chosen. According to the logic behind the calculation of the "$\chi^2$ score", if a feature has a low "$\chi^2$ score", it is independent of the target class, which suggests that it is useless for categorizing data samples.

According to the theory behind chi-square score computation, features with low chi-square scores are independent of the target class and hence useless for categorizing data samples [32]. Chi-square feature selection assesses the independence of events for a collection of data. Using Equation (6), the chi-square feature selection approach assesses

the independence of two events, the occurrence of a feature, and the occurrence of a class by comparing their occurrence rates [32]:

$$x^2 = \frac{(Oserved\ frequency - Expectected\ Frequency)^2}{Expected\ Frequency} \tag{6}$$

## 4. Proposed Approach

In this paper, the proposed approach aims to enhance the accuracy of feature selection using two wrapper methods (PSO and GWO). The popular swarm intelligence algorithm, PSO, operates concurrently with a promising algorithm, GWO, serving as a wrapper method. The main architecture of this phase is shown in Figure 1. In phase 2, another enhancement is added to improve the execution time by implementing a filtering method (chi-square) before this combination. The chi-square filtering method is integrated into the process to eliminate the most irrelevant features, aiming to decrease the execution time of the feature selection process. To ensure the precision of the results, the data were categorized based on three parameters: the dimensions of the features, records, and the number of class attributes before analyzing the results. The first phase of the proposed approach is mainly compared to a hybrid approach that combines a salp swarm algorithm (SSA) with particle swarm optimization (PSO), as well as pure PSO and other pure algorithms such as salp swarm optimization (SSA), a bat algorithm (BAT), and a genetic algorithm (GA). This evaluation was conducted using seven distinct datasets. The subsequent phase of the approach was compared to similar hybrid algorithms that incorporated both filtering and wrapper methods, such algorithms that combine MRMR with PSO (MRMR-PSO), MRMR-SSA, MRMR-GA, ant colony optimization (MRMR-ACO), and ant lion optimization (MRMR-ALO). This comparison was performed using nine different datasets. Finally, both phases of the approach (before and after the addition of a chi-square filter) were compared to the primary PSO algorithm, and their respective advantages and disadvantages were discussed.

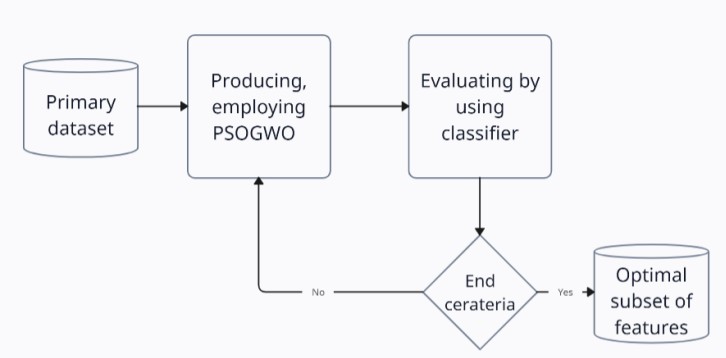

**Figure 1.** Architecture of phase 1 (PSOGWO).

### 4.1. PSOGWO (Phase 1)

In this section, the structure of PSOGWO is outlined, as shown in Figure 1. PSOGWO combines two wrapper algorithms called PSO and GWO. The objective of this phase is to assess the effectiveness of integrating GWO and PSO algorithms that have different search strategies, as shown in [30,31]. PSO is a widely used feature selection algorithm in the literature. With this change, the PSO's update mechanism is integrated into the GWO's main structure. The detailed structure of the PSOGWO approach is given in phase 1 of Figure 2.

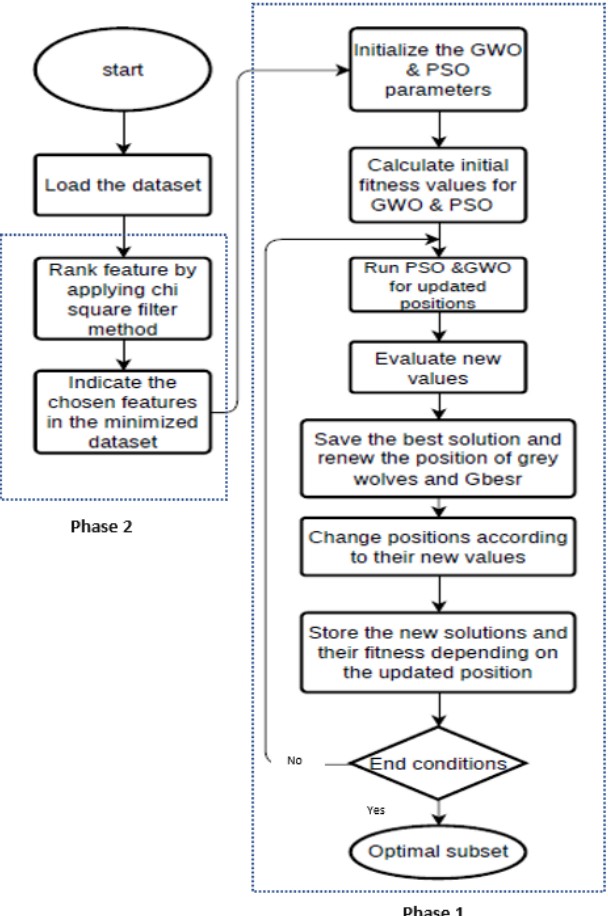

**Figure 2.** Flow chart for chi2-PSOGWO.

In PSOGWO, the first and second steps are to establish the parameters and create a population that represents a collection of potential solutions to a given problem (feature selection). The effectiveness of each solution is then assessed by computing its fitness function and selecting the best one. The PSOGWO algorithm's subsequent stage involves updating the population using the GWO and PSO algorithms, which run simultaneously. The fitness functions of grey wolves and the global best are then compared, and their values are updated. The fitness function is computed using Equation (7) [33]:

$$fitness = weight_{acc} \times accuracy(agent) + weight\_feature \times \frac{tot\_feat - sel\_feat}{tot\_feat} \quad (7)$$

where *tot_feat* is the total number of features contained in the agent *sel_feat*, is the number of features the agent has chosen, and *accuracy(agent)* is the classification accuracy supplied by the agent.

After that, the grey wolves and the global best change their positions according to these new values and the parameters are updated based on the new position. This operation is repeated until the end conditions are met. The result is a vector of ones and zeros that indicates whether a feature was selected or dropped. Phase 1 of the proposed approach is shown in #phase 1 of Algorithm 3.

| **Algorithm 3.** *Chi square-PSOGWO pseudocode*: |
|---|
| 1: **Initialization** dataset |
| #Phase 2 |
| 2: **Rank** features using Chi-square filter method |
| 3: **indicate** the chosen features in the minimized dataset |
| #Phase 1 |
| 4: **initialize** population of particles and velocities |
| 5: **initialize** grey wolf populations |
| 6: **initialize** w, a, A and c values |
| 7: **calculate** the fitness of each search agent and particle |
| 8: **while** t < maximum number of iterations |
| 9:      **for** each particle |
| 10:          **update** velocity by Equation (2) |
|              **update** position of particles by Equation (1) |
| 11:      **end for** |
| 12:      **for** each GWO search agent |
| 13:          update the position of current search agent by Equation (5) |
| 14:      **end for** |
| 15:      **compare fitness** of Gbest, Localbest and $x_\alpha, x\beta, x\delta$ and update Gbest, $x_\alpha, x\beta, x\delta$ by the best values respectively (Gbest, Localbest $= x_\alpha) > x\beta > x\delta$ |
| 16:          **update** A, c, w |
| 17: **End while** |

### 4.2. Chi-Square PSOGWO (Phase 2)

In this section, the integration of PSOGWO with a chi-square filtering algorithm (phase 2) is described, as illustrated in Figure 3. This phase is composed of the following stages:

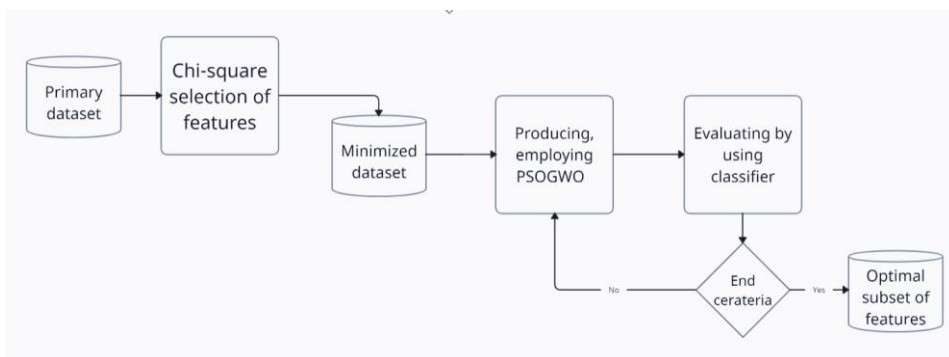

**Figure 3.** Architecture of phase 2 (chi-square-PSOGWO).

The chi-square technique is used to filter the dataset and select only the necessary and relevant features. The chi-square score of each feature is determined.

To enable experts to decide by their own insight which extraneous or additional information to perform, the proposed approach enables the user to handpick the number of selected feature subsets.

After the filtered features are provided, the hybrid wrapper phase (PSOGWO) initializes the search space. This part involves deciding whether to include a feature or discard it. So, the output of this stage will be a string of binary numbers, i.e., one for selected features or one for non-selected features. Figure 2 explains the flowchart of the chi2-PSOGWO approach. Algorithm 3 presents the detailed pseudocode of the proposed chi2-PSOGWO approach. A random forest (RF) classifier was applied to measure the accuracy of the selected features.

## 5. Experiments and Results

In this section, the performances of the suggested phases (PSOGWO and chi2-PSOGWO) are evaluated by comparing the two phases of the proposed approach to other similar algorithms tackling feature selection [1,30,34].

### 5.1. Parameter Settings

Python was used to carry out the proposed hybrid structure's overall implementation. A personal computer (PC) driven by an Intel i5 processor under Windows 10 and with 8 GB RAM was used. Table 1 presents the parameter settings of the proposed approach.

**Table 1.** Parameter settings of algorithms used.

| Parameter settings | |
|---|---|
| Number of search agents | 30 |
| Maximum number of iterations | 100 |
| Number of runs for a single case | 10 |
| $weight_{acc}$ | 0.5 |
| GWO | r1, r2 are random vectors in [0, 1] components of *a* 2 and linearly decrease to 0 over the course of iterations |
| PSO | w starts from 1.0 and decreases by iterations of 1.0—(iter_no/max_iter) r1, r2 are random vectors in [0, 1] c1, c2 = 1 |
| Chi-square | The standard parameters for original chi-square algorithm were used Selection method = numTopFeatures Top features = Experts' handpick |

### 5.2. General Data Settings

According to the literature [1], the feature count ranges from 0 to 19 in the lowest category, from 18 to 46 in the medium category, and 50 and above in the highest category for every attribute selection issue. The number of records is categorized following the same idea of feature categorization approach as that shown in Table 2.

**Table 2.** Summary of categories of used datasets.

| No. of Features | Category | No. of Records | Category |
|---|---|---|---|
| [0, 18] | Lower (L) | [0, 300] | Lower (L) |
| [19, 46] | Medium (M) | [301, 500] | Medium (M) |
| [50, ∞] | Higher (H) | [501, ∞] | Higher (H) |

To prepare the datasets for the task, both real and categorical values are transformed into numeric data. The data are unstructured and have a wide range of values. This variation creates problems in training a model. So, a minimal maximum scaling strategy is then used. A MinMaxScaler normalization technique is used to ensure that the scales of all the data in the database are similar by bringing them all to a common range. All data values can be scaled to have values between 0 and 1 using the MinMaxScaler normalization technique. The normalization technique used by MinMaxScaler is indicated by Equations (8) and (9).

$$Xstd = (X - X.min)\,(X.max - X.min) \tag{8}$$

$$Xscaled = Xstd \, * \, (X.max - X.min) + X.min \tag{9}$$

The minimum and maximum values for the feature X under consideration are denoted by the terms *min* and *max* in Equations (8) and (9). For a specific feature, the normalized values are provided by Equations (8) and (9). All of the values in the datasets are fitted and transformed before being utilized for training and testing [35].

Each dataset is split into test and training instances using the 70–30 rule (70% of training instances, 30% of testing instances).

### 5.3. Experiment 1

The proposed PSOGWO approach is compared to another hybrid algorithm called the SSAPSO algorithm. SSAPSO is an integration of particle swarm intelligence (PSO) and salp swarm intelligence algorithm (SSA). It is also compared to the pure PSO algorithm and other pure algorithms like the BAT and GA algorithms [30,34].

### 5.3.1. Dataset Settings

A set of UCI datasets is utilized to test the first phase [36]. Table 3 shows the seven datasets used in this experiment.

**Table 3.** PSOGWO dataset discerption.

| NO. | Dataset | Features | Records | Class Variables |
|---|---|---|---|---|
| 1. | Ionosphere | 34 | 351 | 2 |
| 2. | Hepatitis | 19 | 155 | 2 |
| 3. | Heart | 13 | 270 | 2 |
| 4. | Breast cancer | 9 | 683 | 2 |
| 5. | Sonar | 60 | 208 | 2 |
| 6. | Lymphography | 19 | 148 | 4 |
| 7. | Waveform | 21 | 5000 | 3 |

### 5.3.2. Results and Discussion

The performance of PSOGWO is assessed by passing the selected features to the classifier to determine the accuracy of feature selection. To achieve a more accurate comparison with competing algorithms, identical datasets and classifiers (KNN) of the competitors' algorithms were used.

As demonstrated in Figure 4, the highest accuracy measure was achieved by PSOGWO in the ionosphere, hepatitis, and heart datasets, which are in the medium (M) and low (L) categories in terms of their number of features and records, respectively. However, its accuracy decreases in the breast cancer dataset, which has a higher number of records, and the sonar dataset, which has a higher number of features. Table 4 explains why the PSOGWO phase appears in the first stage in three datasets and in the second stage in one of the seven datasets that were utilized. It performs better with a medium number of records, lower features, and lower target (no. of classes). Throughout this experiment, it can also be observed that phase 1 performs better in a lower number of classes.

**Table 4.** The results of the accuracy (%) evaluation of feature selection in all datasets.

| No. | Dataset | SSA | SSAPSO | GA | BAT | PSO | PSOGWO | Notes | | |
|---|---|---|---|---|---|---|---|---|---|---|
| | | | | | | | | Feature | Records | Classes |
| **1.** | **Ionosphere** | 93.9 | **95.1** | 91.6 | 90.4 | 94.9 | **95.3** | **M** | **M** | **2** |
| **2.** | **Hepatitis** | 71.7 | 74.8 | 64.3 | 62.9 | 72.1 | **85.3** | **M** | **L** | **2** |
| **3.** | **Waveform** | 78.49 | 79.13 | 78.38 | 79.29 | **79.47** | **87.3** | **M** | **H** | **3** |
| **4.** | **Heart** | 82.3 | **84.7** | 79.0 | 76.1 | 83.1 | **83.3** | L | L | 2 |
| **5.** | **Breast cancer** | 96.3 | **97.8** | 97.5 | 97.2 | **97.6** | 74.3 | L | H | 2 |
| **6.** | **Sonar** | 94.43 | 96.20 | **96.70** | 95.27 | **96.94** | 87.4 | H | L | 2 |
| **7.** | **Lymphography** | 94.43 | 90.20 | **96.70** | 95.27 | **97.6** | 88.2 | M | L | 4 |

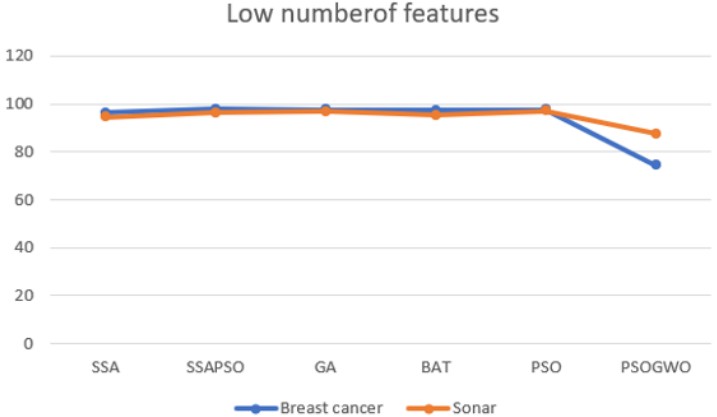

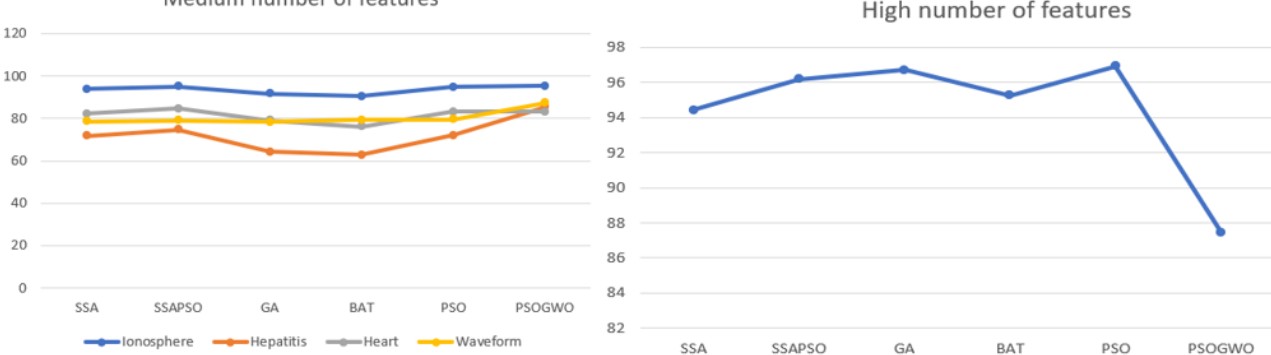

**Figure 4.** Comparing accuracy of feature selection.

*5.4. Experiment 2*

This experiment compares the results of the chi2-PSOGWO approach against MRMR-PSO, MRMR-SSA, MRMR-GA, MRMR-ALO, and MRMR-ACO algorithms explained in [1]. In the initial stage of the proposed configured model, the chi-squared algorithm is parameterized in terms of counts of features to identify the most important features from the various features in the datasets. Consequently, the obtained features of the first stage are evaluated by the second stage (PSOGWO) in order to reach the final subset of features.

5.4.1. Dataset Settings

The suggested chi2-PSOGWO hybrid approach is evaluated through a number of trials using different Kaggle datasets [37]. Table 5 gives an overview of the datasets that were used. The name of the dataset, the number of features or attributes, the record number in each dataset, and the class variables to which each dataset belongs are the four different types of descriptions that are included in the table.

**Table 5.** Chi2-PSOGWO dataset discerption.

| NO. | Dataset | Records | Features | Class Variables | Filter Selection |
|---|---|---|---|---|---|
| 1. | Abalone | 4177 | 8 | 3 | 6 |
| 2. | Breast cancer | 569 | 30 | 2 | 15 |
| 3. | Banknote authentication | 1372 | 5 | 2 | 4 |
| 4. | Car evaluation | 1727 | 6 | 4 | 5 |
| 5. | Heart disease | 303 | 13 | 2 | 9 |
| 6. | Habitats | 142 | 19 | 2 | 10 |
| 7. | Iris | 150 | 4 | 3 | 3 |
| 8. | Lymphography | 148 | 18 | 4 | 10 |
| 9. | Wine | 178 | 13 | 3 | 10 |

5.4.2. Result and Discussion

This section examines the outcomes of the hybrid chi2-PSOGWO approach in comparison with other hybrid filter–wrapper approaches, such as MRMR-SSA, MRMR-PSO, MRMR-GA, MRMR-ACO, and MRMR-ALO [1]. The evaluation is based on their accuracy (in %) assessed by the RF classifier and uses the dataset group of the competitor algorithms. as depicted in Table 5.

An analysis presented in Table 6 shows that chi2-PSOGWO is the most successful in the breast cancer, heart disease, habitats, and wine datasets. Additionally, it showed better performance when the number of features and records was medium (M) or low (L). However, when the number of records increased (e.g., in the banknote authentication dataset), accuracy diminished, even when the number of records was high. Also, it operates more effectively with lower class attributes. As demonstrated in Figure 5, the suggested approach performs better with a medium dimension of features. It also performs better with a lower number of records.

**Table 6.** Comparing accuracy using RF classifier (%).

| NO. | Dataset | MRMR-SSA | MRMR-GA | MRMR-ALO | MRMR-ACO | MRMR-PSO | Chi2-PSOGWO | Notes | | |
|-----|---------|----------|---------|----------|----------|----------|-------------|-------|-------|------------------|
| | | | | | | | | Features | Records | Class Attributes |
| 1 | Abalone | **78.87** | **77.73** | 54.33 | 76.57 | 56.38 | 53.54 | L | H | 3 |
| 2. | Breast cancer | 93.69 | 89.08 | 92.72 | 93.89 | 91.83 | **97.24** | **M** | **H** | 2 |
| 3. | Banknote authentication | 95.63 | 88.17 | **94.42** | **96.12** | **94.42** | 87.10 | L | H | 2 |
| 4. | Car evaluation | 73.87 | 73.33 | 70.22 | **74.82** | 70.71 | 61.01 | L | H | 4 |
| 5. | Heart disease | **82.98** | 80.67 | 84.44 | 82.98 | 81.02 | **95.961** | **L** | **M** | 2 |
| 6. | Habitats | 88.53 | 80.09 | 84.68 | **89.71** | 88.98 | **92.5** | **M** | **L** | 2 |
| 7. | Iris | 94.42 | 94.42 | **96.84** | 87.73 | **96.60** | 95 | L | L | 3 |
| 8. | Lymphography | **86.89** | 83.33 | 80.00 | 82.22 | **84.95** | 81.33 | M | L | 4 |
| 9. | Wine | 95.63 | **95.72** | 86.89 | 92.72 | 92.72 | **99.44** | **L** | **L** | **3** |

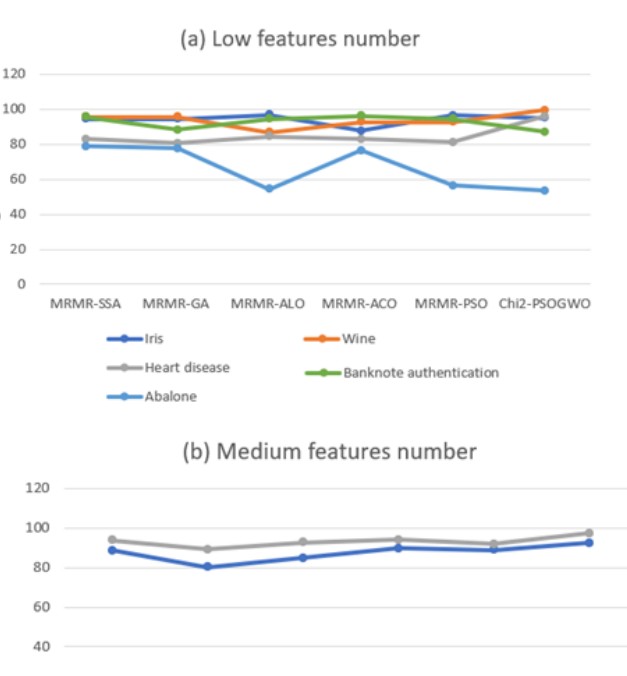

**Figure 5.** Comparing accuracy of chi2-PSOGWO.

*5.5. Experiment 3*

In this section, the two phases of the approach proposed in this paper, chi2-PSOGWO and PSOGWO, are compared to the main PSO algorithm in terms of accuracy. The goal is to evaluate the improvement achieved by combining the main particle swarm intelligence algorithms with other wrapper-based methods (GWO) and filter-based methods (chi-square). Execution time is used to determine if incorporating the filter-based algorithm (chi-square) affects the performance of the PSOGWO combination.

5.5.1. Dataset Settings

The same dataset group (Table 2) as in experiment 1 is used to compare the two phases of the proposed approach.

5.5.2. Results and Discussion

The effectiveness of the PSO and GWO combination can be showcased through experiments carried out on datasets of diverse dimensions, record sizes, and class attributes. The experiments, as depicted in Figure 6, reveal a noticeable enhancement in the accuracy of PSO when integrated with GWO in datasets such as ionosphere, hepatitis, heart, waveform, and lymphography. Furthermore, in the case of the breast cancer, hepatitis, and waveform datasets, chi2-PSOGWO is involved in the second stage. As shown in Table 7, the two phases of the proposed approach show an improved performance in categories with a medium number of features, with no noticeable impact on the number of records or class attributes. Table 8 demonstrates the execution time of both PSOGWO and chi2-PSOGWO across all the datasets. Figure 7 presents a comparison of the execution time in the two phases of the proposed approach. The chi2-PSOGWO approach exhibits a reduced execution time compared to the PSOGWO method in the majority of datasets. This is attributed to the filtering performed by the chi-square algorithm, which effectively reduces the number of features entering the subsequent stage and thereby minimizes computational time.

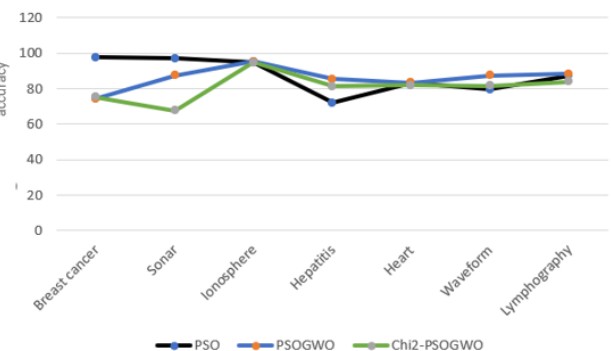

**Figure 6.** Comparing accuracy of PSO, PSOGWO, and chi2-PSOGWO.

**Table 7.** Comparing accuracy of PSO, PSOGWO, and chi2-PSOGWO.

| Datasets | PSO | PSOGWO | Chi2-PSOGWO | Features | Records | Classes |
|---|---|---|---|---|---|---|
| Breast cancer | **97.6** | **74.3** | **75.3** | **L** | **L** | **2** |
| Sonar | **96.94** | **87.4** | **67.7** | **L** | **H** | **2** |
| Ionosphere | **94.9** | **95.3** | **94.7** | **M** | **M** | **2** |
| Hepatitis | **72.1** | **85.3** | **81.3** | **M** | **M** | **2** |
| Heart | **83.1** | **83.3** | **82** | **M** | **L** | **2** |
| Waveform | **79.47** | **87.3** | **81.72** | **M** | **H** | **3** |
| Lymphography | **87.425** | **88.2** | **84** | **M** | **L** | **4** |

**Table 8.** Comparing execution time of PSOGWO and chi2-PSOGWO.

| Datasets | PSOGWO | Chi2-PSOGWO | Features | Records | Classes |
|---|---|---|---|---|---|
| Ionosphere | 649.55 | 412.97 | M | M | 2 |
| Hepatitis | 65.5 | 20.8 | M | M | 2 |
| Heart | 12.77 | 9.2 | M | L | 2 |
| Breast cancer | 16.9 | 19.7 | L | L | 2 |
| Sonar | 17.11 | 16.7 | L | H | 2 |
| Lymphography | 55.946 | 14.49 | M | L | 4 |
| Waveform | 649.55 | 412.965 | M | H | 3 |

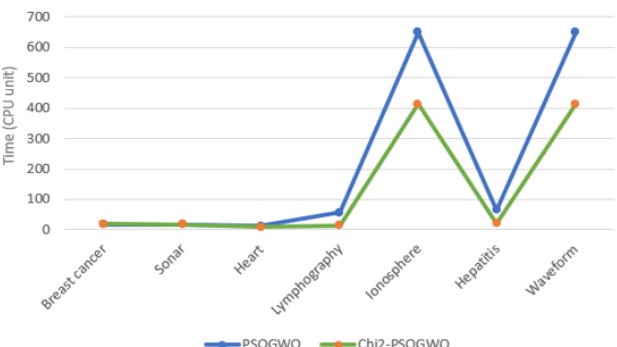

**Figure 7.** Comparing execution time of PSOGWO and chi2-PSOGWO.

## 6. Summary and Observations

The proposed approach entails assessing the accuracy and execution time of the PSO algorithm subsequent to its hybridization with both a wrapper method (GWO) and a filter method (chi-square). The performance of this hybrid approach was then compared with other pure and hybrid algorithms to determine its effectiveness. The datasets were categorized based on their parameters, such as the number of features, records, and class attributes (Tables 3 and 5). The dataset was preprocessed by converting the non-numeric data into numeric data; any null values were removed to ensure accurate performance and the Minimum–Maximum Scaler was applied to scale the records to a common range. To ensure a precise comparison, the same classifiers (RF, KNN) were employed across the compared algorithms. The proposed approach excels in two phases: surpassing the pure PSO algorithm (Figure 6, Table 7) and outperforming other significant pure and hybrid algorithms in terms of accuracy (Tables 4 and 6, Figures 4 and 5). It can be observed that a learning algorithm's running time may be significantly decreased by considerably reducing the number of redundant features, which aids in understanding the fundamental complexities of a practical classification problem (Figure 8). Therefore, the reason that the chi2-PSOGWO approach has lower accuracy than PSOGWO could be attributed to the fact that it may overlook some important features. Another possible explanation is that the fitness function uses the ratio between the difference between the selected features and the original features to the original feature set. When a smaller (filtered) dataset is used, it is possible for this ratio to decrease. This decrease in ratio could result in a lower theoretical accuracy compared to the situation where no filters are applied, as demonstrated in Equation (7).

However, the accuracy measured by chi2-PSOGWO was acceptable for effectively reducing running time and maintaining a high prediction accuracy, as shown in Figure 7. As a conclusion, the results were very encouraging in terms of achieving superior performance and outperforming benchmark algorithms in many cases compared to other similar approaches.

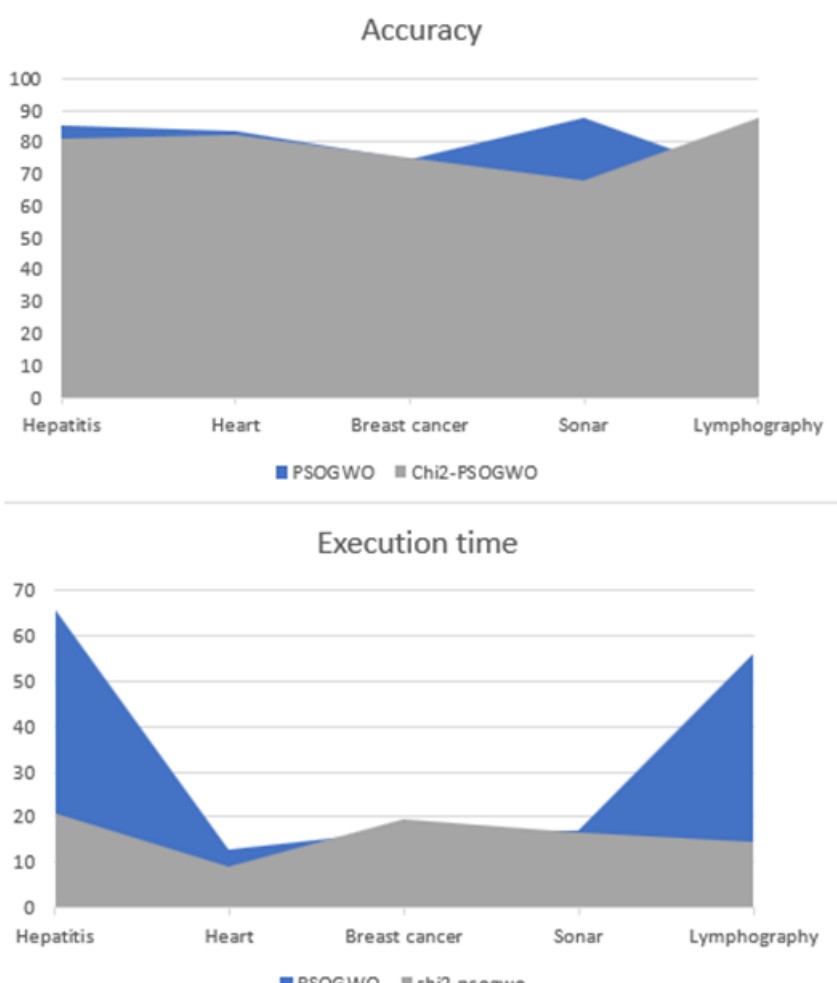

**Figure 8.** Evaluation of accuracy and execution time.

## 7. Conclusions and Future Work

This paper has the purpose of effectively tackling the challenges associated with feature selection and presents an approach that effectively resolves these issues. Various difficulties related to feature selection, such as classification accuracy and execution time, have been thoroughly evaluated. The proposed approach was implemented in two distinct phases, with each phase utilizing a unique dataset to achieve the best practice. Based on the experimental data, the proposed approach demonstrated its superiority over existing feature selection techniques. A comparative analysis was conducted between the two phases of the proposed approach and several established methods. The experimental results further confirmed that both the chi2-PSOGWO and PSOGWO algorithms demonstrated a noticeable enhancement in accuracy compared to other hybrid techniques. However, the chi2-PSOGWO algorithm exhibited a superior improvement in both accuracy and execution time. This research holds the potential for further improvement and application in various domains, including multi-objective problems, engineering design, parameter estimation, text clustering, text summarization, text categorization, image segmentation, mathematical benchmark functions, and other feature selection applications.

**Author Contributions:** Conceptualization, A.A. and R.M.; methodology, L.A.-H.; software, R.M.; validation, A.A., R.M. and L.A.-H.; formal analysis, L.A.-H.; investigation, R.M.; resources, L.A.-H.; data curation, R.M.; writing—original draft preparation, R.M.; writing—review and editing, L.A.-H.; visualization, R.M.; supervision, A.A.; project administration, A.A.; funding acquisition, R.M. All authors have read and agreed to the published version of the manuscript.

**Funding:** This research received no external funding.

**Data Availability Statement:** The experiments were conducted over two groups of datasets. The datasets are provided on the Kaggle repository (www.kaggle.com), accessed on 1 April 2022 and UCI repository (https://archive.ics.uci.edu/mL/datasets.php), accessed on 1 May 2022.

**Conflicts of Interest:** The authors declare no conflict of interest.

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
