# Peer review of "An Optimized Hybrid Approach for Feature Selection Based on Chi-Square and Particle Swarm Optimization Algorithms"

_data_

Round 1
Reviewer 1 Report
Comments and Suggestions for Authors
This article proposes a feature selection method that combines Grey Wolf algorithm and Particle Swarm Optimization algorithm to improve the accuracy and efficiency of feature selection.
1. The author should explain why the two algorithms are combined. Is this combination specifically effective for feature selection problems, or is it effective for general optimization problems?
2. Figure 3 shows the two stages of the algorithm. Why is the process from stage 2 to stage 1?
3. Page7:“This paper outlines a two-phase structure to address the feature selection issue as illus- 236 trated in Fig 1.” Here Fig 1should be Figure 3.
4. Table 1 lists the various parameters of the algorithm. Are these parameters all optimal? Do these settings have an impact on the performance of the algorithm? Especially, the setting of weights.
Comments on the Quality of English LanguageThere are some minor errors in some areas, please carefully check.
Author Response
Dear reviewer,
We are grateful for the valuable suggestions and feedback on the manuscript titled “An Optimized Hybrid Approach for Feature Selection based on Chi-square and PSO Algorithms”.
Here is a point-by-point response to the comments and concerns.
- Comment 1: [The author should explain why the two algorithms are combined. Is this combination specifically effective for feature selection problems, or is it effective for general optimization problems?]
Response: Thank you for pointing this out. We agree with this comment. The objective of the proposed approach is to improve the accuracy of feature selection by incorporating two wrapper methods, Particle Swarm Optimization (PSO) and Grey Wolf Optimization (GWO). PSO works in parallel with GWO, which acts as a wrapper method. Additionally, to further enhance the efficiency of the approach, a filtering method (Chi-Square) is applied prior to this combined process to reduce execution time. Therefore, we have made some changes in the introduction section [line no.64 : line no.80] and proposed approach section [line no.238 : line no. 254] that clarify each algorithm and clarify the reason for implementing the in this way.
- Comment 2: [Figure 3 shows the two stages of the algorithm. Why is the process from stage 2 to stage 1?]
Response: The experiments were done in two phases. First phase is running both PSO and GWO algorithms in parallel. Then in second phase the Chi-square filtering algorithm is added before running the parallel combination aiming to improve the execution time by removing the most irrelevant features.
We agree that Figure 3 may occur some confusion. We have, accordingly, removed this figure and added a separated architecture for each phase that can be shown now on Figure 3 and Figure 6 [line no. 294 and line no. 318] to emphasize this point.
- Comment 3: [ Page7:“This paper outlines a two-phase structure to address the feature selection issue as illustrated in Fig 1.” Here Fig 1should be Figure 3.]
Response: Agree. We have, accordingly, modified the text to emphasize this point.
- Comment 4: [ Table 1 lists the various parameters of the algorithm. Are these parameters all optimal? Do these settings have an impact on the performance of the algorithm? Especially, the setting of weights.]
Response: The parameters of the used algorithms PSO,GWO Chi-square were optimal. The other parameters like Number of search agents, Maximum number of iterations, Number of runs for a single case were set to the same values of the competitor algorithms to guarantee a fair comparison. weight_acc is a required parameter for the functions. Its value was set to 0.5 during all the trials to get an accurate result.
In addition to the above comments, all spelling and grammatical errors pointed out by the reviewers have been corrected.
We look forward to hearing from you in due time regarding our submission and to respond to any further questions and comments you may have.
Best regards,

Reviewer 2 Report
Comments and Suggestions for Authors
Greetings,
No issues found abstract,
Introduction: the first two paragraphs can be omitted. The authors have also given an overview of the basic techniques that can be removed as well. There is little relevance to them.
In order to provide a comprehensive understanding of the effectiveness and performance of this hybrid approach, the implementation details, hyperparameter settings, and evaluation metrics would have to be specified in the research paper or project. To assess the feasibility of the approach, it should also be empirically validated using relevant datasets.
Comments on the Quality of English Language
Minor editing of English language required
Author Response
Dear reviewer,
We are grateful for the valuable suggestions and feedback on the manuscript titled “An Optimized Hybrid Approach for Feature Selection based on Chi-square and PSO Algorithms”.
Here is a point-by-point response to the comments and concerns.
- Comment 1: [ Introduction: the first two paragraphs can be omitted. The authors have also given an overview of the basic techniques that can be removed as well. There is little relevance to them]
Response: Thank you for pointing this out. We agree with this comment. Therefore, we have removed the first two paragraphs and made some edits into the introduction part [line no.64 : line no.79] that make it more relevant to the topic of paper.
- Comment 2: [ The implementation details, hyperparameter settings, and evaluation metrics would have to be specified in the research paper or project.]
Response: Agree. We have, accordingly, added the software and hardware details [line no.334 to 336] to emphasize this point. The hyperparameter and evaluation metrics have been discussed in the proposed approach section [line no.238 : line no. 254]. The details of hyperparameter can be found on Table 2.
- Comment 3: [To assess the feasibility of the approach, it should also be empirically validated using relevant datasets]
Response: Thank you for this suggestion. The two phases of the approach are assessed by two groups of datasets that can be shown in Table 3 and Table 5.
In addition to the above comments, all spelling and grammatical errors pointed out by the reviewer have been corrected.
We look forward to hearing from you in due time regarding our submission and to respond to any further questions and comments you may have.
Best regards,

Reviewer 3 Report
Comments and Suggestions for Authors
Dear authors,
This article presents a feature selection method using a hybrid particle swarm optimization approach. The method is based on two phases to improve feature selection accuracy and system execution time. The paper is interesting and relevant. However, it is recommended to go through some review comments before acceptance as follows:
1- The selection of hybrid parameters is very important for comparison, therefore, on what basis the hybrid parameters were selected in Table (1)? Please discuss!
2- For experiment 1, the authors have compared their method with the SSA, SSAPSO, GA, PSO, and BAT. Some of these algorithms are old and are not very significant for comparison. It is recommended to add two more recent algorithms for comparison purposes.
3- For experiment 2, other algorithms were used, why such inconsistency in compassion?
4- The authors can explicitly mention all the hyperparameters used for the compared algorithm in one table.
5-Please make sure that all the abbreviations are carefully defined before the first use of the manuscript., especially the names of the algorithms.
6- The conclusion should be more carefully rewritten, summarizing what has been learned and why it is interesting and useful.
7- What are the differences between the developed PSOGWO variants and some recent PSO variants such as the improved salp swarm algorithm based on particle swarm optimization, the combined social engineering particle swarm optimization, the self-adaptive gradient-based particle swarm optimization algorithm, or the novel oppositional unified particle swarm gradient-based optimizer? Please discuss this in the introduction section!
Best wishes to the authors!
Author Response
Dear reviewer,
We are grateful for the valuable suggestions and feedback on the manuscript titled “An Optimized Hybrid Approach for Feature Selection based on Chi-square and PSO Algorithms”.
Here is a point-by-point response to the comments and concerns.
- Comment 1: [ The selection of hybrid parameters is very important for comparison, therefore, on what basis the hybrid parameters were selected in Table (1)? Please discuss!]
Response: Thank you for this suggestion. It have already been mentioned in [line no, 336] that Table 1 includes the parameters used in the purposed hybrid approach.
- Comment 2: [ For experiment 1, the authors have compared their method with the SSA, SSAPSO, GA, PSO, and Some of these algorithms are old and are not very significant for comparison. It is recommended to add two more recent algorithms for comparison purposes.]
Response: We appreciate your suggestion. It would have been interesting to explore this aspect. However, in the case of our study, this issue was targeted in experiment 1; the hybrid wrapper phase (PSOGWO) is compared to a similar hybrid wrapper algorithm (SSAPSO) to assess its performance. Additionally, we compared it to pure algorithms such as PSO, SSA, GA, and BAT to highlight the improvements achieved through hybridizing wrapper algorithms. Some changes have been added to the proposed approach section [line no.238 : line no. 254] to clarify that.
- Comment 3: [ For experiment 2, other algorithms were used, why such inconsistency in compassion? ]
Response: You have raised an important point here. However, we believe that it would be more appropriate to compare each phase of the purposed approach to similar algorithms to assess the performance of them. For example, the PSOGWO is mainly compared to a hybrid wrapper algorithm (SSAPSO). For the second phase (Chi2-PSOGWO) a filtering algorithm (Chi-Square) is added before the wrapper phase. So that, this phase has been compared to other filter-wrapper approaches like MRMR-PSO,..etc. Some changes have been added to the proposed approach section [line no.238 : line no. 254] to clarify that.
Comment 4: [ The authors can explicitly mention all the hyperparameters used for the compared algorithm in one table.]
Response: Agree. We have, accordingly, discussed the hyperparameter [line no.240 to 242] to emphasize this point. The details of hyperparameter can be found on Table 2.
- Comment 5: [ Please make sure that all the abbreviations are carefully defined before the first use of the manuscript., especially the names of the algorithms.]
Response: Thank you for pointing this out. We agree with this comment. Therefore, we have conducted a thorough review of the abbreviations used and made the necessary edits accordingly.
- Comment 6: [ The conclusion should be more carefully rewritten, summarizing what has been learned and why it is interesting and useful.]
Response: Agree. We have, accordingly, carefully revised and modified the conclusion to effectively summarize the work in an engaging manner [ Line no. 513 to 529].
- Comment 7: [ What are the differences between the develop d PSOGWO variants and some recent PSO variants such as the improved salp swarm algorithm based on particle swarm optimization, the combined social engineering particle swarm optimization, the self-adaptive gradient-based particle swarm optimization algorithm, or the novel oppositional unified particle swarm gradient-based optimizer? Please discuss this in the introduction section! ]
Response: We appreciate your recommendation. Experiment 1 has already conducted a comprehensive comparison between the PSOGWO phase and the improved salp swarm algorithm based on particle swarm [Line to363 to line 395 ]
In addition to the above comments, all spelling and grammatical errors pointed out by the reviewers have been corrected.
We look forward to hearing from you in due time regarding our submission and to respond to any further questions and comments you may have.
Best regards,

Reviewer 4 Report
Comments and Suggestions for Authors
The authors presented the results of a comparative analysis of the efficiency of algorithms for optimal feature selection. The authors propose to combine two optimization algorithms: gray wolf (GWO) and particle swarm (PSO). In addition, the novelty of the study is the combination of these methods with pre-filtering of data using the Chi-square method. The efficiency of the combined algorithm with and without pre-filtering is compared by the authors with the basic PSO algorithm. The study as a whole was performed correctly. A variety of data sets were used to evaluate the performance of the algorithms.
A few suggestions for improving the manuscript.
1. The introduction and literature review do not give a clear understanding why the authors chose these methods for analysis. I propose to justify this choice in more detail. A description of the limitations of the study and/or a more detailed presentation of future research may also be a suitable solution.
2. The authors presented the comparative efficiency of the studied algorithms. However, I propose to show estimates of the speed of algorithms in absolute units. In this case, it is necessary to detail the characteristics of the software and hardware used.
3. Y-axes need to be labeled on all charts.
4. The sentence on line 484 does not end with a period.
Author Response
Dear reviewer,
We are grateful for the valuable suggestions and feedback on the manuscript titled “An Optimized Hybrid Approach for Feature Selection based on Chi-square and PSO Algorithms”.
Here is a point-by-point response to the comments and concerns.
- Comment 1: [ The introduction and literature review do not give a clear understanding why the authors chose these methods for analysis. I propose to justify this choice in more detail. A description of the limitations of the study]
Response: Thank you for pointing this out. We agree with this comment. Therefore, we have made some changes in the introduction section [line no.64 : line no.79] and proposed approach [line no.238 : line no. 254] that clarify each algorithm and clarify the reason for implementing the in this way. The limitation of the purposed approach is discussed in section 6 [specially in line no. 499 to 510]
- Comment 2: [ The authors presented the comparative efficiency of the studied algorithms. However, I propose to show estimates of the speed of algorithms in absolute units. In this case, it is necessary to detail the characteristics of the software and hardware used.]
Response: Agree. We have, accordingly, modified the parameter by adding the software and hardware details [line no.334 to 336] to emphasize this point.
- Comment 3: [ Y-axes need to be labeled on all charts.]
Response: We agree with this and have edited the Y- axis of all charts throughout the manuscript.
- Comment 4: [ The sentence on line 484 does not end with a period.]
Response: Thank you for this suggestion. It has been edited.
In addition to the above comments, all spelling and grammatical errors pointed out have been corrected.
We look forward to hearing from you in due time regarding our submission and to respond to any further questions and comments you may have.
Best regards,

Round 2
Reviewer 3 Report
Comments and Suggestions for Authors
The paper can be accepted for publication